# Social and nutritional factors controlling the growth of honey bee (*Apis mellifera*) queens

Omer Kama[1,2], Hagai Yehoshua Shpigler [1]*

1 Department of Entomology, Agricultural Research Organization, The Volcani Institute, Rishon LeZion, Israel, 2 The Robert H. Smith Faculty of Agriculture, Food and Environment, The Hebrew University in Jerusalem, Rehovot, Israel

* hagais@volcani.agri.gov.il

## Abstract

The honey bee queen is essential for colony function, laying hundreds of eggs daily and determining the colony's genetic composition. Beekeepers cultivate and trade queens to enhance colony health and productivity. Despite its significance, artificial queen rearing in foster queenless colonies has remained largely unchanged for over a century, offering limited control over the environmental conditions influencing larval development. In this study, we developed a laboratory-based method for queen bee rearing, establishing a protocol for rearing queens in cages by nurse bees in the lab under controlled environmental conditions. We first investigated the minimal number of worker bees required to rear a single queen and found that groups of 200 workers raise queens with comparable success and weight to those reared in foster colony. As a proof of concept, we examined the impact of larval age on rearing success in our new system. We found that younger larvae developed into heavier and larger queens than older larvae, as recorded in the past using the traditional rearing method. Additionally, we assessed the influence of pollen nutrition on queen-rearing success, finding that a high pollen concentration is crucial for optimal queen development. These findings and the new method provide a foundation for studying queen bee-rearing behavior and development in the lab. We expect that it will be used to uncover factors that impact this important process in honey bee biology.

## Introduction

The honey bee (*Apis mellifera*) queen is the most important individual within the colony, as the reproductive success of the colony depends on her viability [1]. Beekeepers typically replace their queens at least once every other year to maintain colony vigor [2]. To meet this demand, new queens are raised annually. Artificial queen rearing was developed in the late 19th century in the U.S. by Gilbert Doolittle [3,4], shortly after the establishment of the modern Langstroth method of beekeeping. This process is based on the principle that every diploid egg has the potential to develop into either a queen or a worker, depending on the social and physical environment of the developing larva [5]. A new queen can be artificially reared by transferring young larvae from worker cells into large, vertically oriented queen cells in queenless colonies a process known as grafting [3,6]. Queen breeders, use specialized equipment and knowledge of honey bee biology to produce large number of queens.

**Data availability statement:** All relevant data are within the manuscript and its Supporting Information files.

**Funding:** The author(s) received no specific funding for this work.

**Competing interests:** The authors have declared that no competing interests exist.

The queen-rearing industry includes thousands of breeders who produce millions of queens annually [7,8].

The honey bee queen has unique morphology, physiology, and behavior that support her tasks performance in the hive with maximal efficiency [1]. Throughout her adult life cycle, the queen's behavior and physiology adapt to meet the demands of each stage. In the early stages of her life, after emerging from the pupa, the queen displays aggressive behavior towards rival virgin queens, coinciding with the development of her venom sac and stinger. During her second week, the virgin queen embarks on nuptial flights, during which her flight muscles and navigation abilities fully develop [9,10]. Once the queen returns from her mating flights, she remains in the hive, dedicating most of her time to egg-laying, venturing outside only during swarming events [11,12]. At the reproductive stage, her ovaries are fully developed, while her flight abilities and venom sac diminish. The success of a high-quality queen throughout all her adult life stages depends significantly on her genetic background and the environment in which she was raised [13,14].

Honey bee colonies will rear new queens under three primary circumstances: in spring before swarming, as a replacement for a malfunctioning queen, or at emergency when the queen of the colony unexpectedly dies [1]. The queen's larval development time is shorter than worker bees [12]. Early emergence from the pupa is advantageous as it minimizes the period the colony is queenless, which is crucial for maintaining the colony's reproductive potential [15]. Moreover, Competition among young queens is intense; the first queen to emerge typically attacks and kills rival queen pupae by stinging them before they can emerge [1]. Queen breeders often raise multiple queens simultaneously within a single colony. To protect the pupae from the aggression of other queens, breeders separate the pupae around day ten of their development. A foster colony used for queen rearing typically contains tens of thousands of workers, frames with pupae, no young larvae, and no queen [16]. The colony is well-nourished with honey and pollen to support the growth of queen larvae. Queen breeders introduce 40-60 young larvae to these colonies in special cells process known as grafting [1,3]. The foster colony will then rear the larvae into queens as an emergency response. There are several methods to build the queen builders colonies. It can be queenless or queen right, in small hives or full hives [6]. To ensure the production of high-quality queens, breeders carefully construct the best possible foster colonies and select larvae from highly productive colonies. These methods are highly efficient and meets market demands. Nevertheless, the breeders have only limited control over the nutrition that the colony gathers, and the foster colonies are exposed to environmental factors such as weather changes, predators, parasites, pesticides, and diseases including viruses [16]. As a result, the rearing process sometimes fail. This raises the question of whether it is possible to rear queens in a more controlled environment that enhance the success and desired outcomes of the breeding process.

The quality of a honey bee queen is influenced by both internal and external factors. Internal factors include the genetic background of the larvae and the genetic source of the workers that rear the queen [14]. External factors include the age of the larvae, the type of nutrition provided, and the strength of the foster colony [3,14]. Former studies shows that younger larvae, develop into larger and more fertile queens compared to older larvae [17]. However, the control that a queen breeder has over the growth process is limited and generally ends after the larvae are introduced to the foster colony. In contrast, bumble bee breeding is conducted indoors, where conditions are tightly controlled to maximize success [18–20]. This indoor breeding approach frees bumble bee breeders from the constraints of seasonality, enabling the year-round production of pollinators. However, this approach is not feasible for honey bees, since they live in large perennial colonies that cannot thrive indoors. This limitation reduces the ability of breeders to control environmental conditions, nutrition, and exposure to

pathogens during the honey bee queen's development. Controlling the rearing conditions of honey bee queens may be useful to improve their quality.

An alternative approach to honey bee growth in the laboratory is in vitro rearing by hand [21–23]. In vitro larval rearing of honey bees is a well-established technique used for various purposes, including research on nutrition and risk assessment of insecticides [24–26]. Depending on the feeding protocol, in vitro rearing can be used to produce either workers or queens [27]. This method is particularly valuable for studying the development and genetics of honey bees. However, since the approach is artificial, it is not suitable for investigating the natural behavior of workers and the queen larvae during the queen-rearing process

In the current study, we introduce a new method for queen rearing under controlled environmental conditions using groups of nurse bees housed in cages under controlled lab conditions. To each group we introduce a single larva in a queen cup for queen rearing and we tracked the acceptance rate of the larvae by bees and the weight of the developing pupae. This method allows us to study the natural process of queen growth in repeatable and reliable measure. Our first experiment aimed to determine the minimal number of nurse bees required to rear a queen. After establishing the method we conducted two experiments as a proof of concept, we tested the effect of larval age on the development of queens within our system in compared to former data [28–30]. Second, we explored the impact of pollen nutrition on queen rearing. Our findings establish the method and demonstrate its use to study basic questions on factors that impact queen rearing behavior and queen growth in the lab.

## Materials and methods

### Honey bees

Honey bees from the apiary located at the Volcani Institute in Israel were used for the study. The colonies were treated followed the regular procedure in Israel. Ten colonies were dedicated to the study and in each experiment, we used day old bees from two colonies. The experiments were conducted in the summer of 2023 and spring of 2024. No permits were required to access the hives or collect the bees and the larvae.

Frames with emerging pupae were collected from the colonies and placed in a six-frame hive box in an acclimated room (34°C ± 1, RH = 60% ± 5). The next day emerging bees were collected into a plastic container and transferred to cages. The number of bees in each cage was estimated by weight using a scale with an accuracy of 0.01 g. We used queen monitoring cages for this study [31]. The cage size is 10 × 15 × 5 cm, with the back of a plastic frame. Each cage has four round holes designed to fit 5 ml tubes (diameter – 9 mm), two on the sides and two at the top. These holes are used for feeding and for the introduction of larval queen cups. The cages were supplied with pollen paste made of 90% pollen and 10% sugar water (60% sugar in the water), a 5 ml tube of honey, and a 5 ml tube of water. The food was refreshed every other day. The cages were kept in an acclimated room with conditions resembling a normal hive (Temp: 34° ± 1°, RH: 60% ± 5%) throughout the experiment.

### Queen rearing

Frames with young larvae were collected from colonies from the Volcani apiary. Day-old larvae were collected from the cells using a grafting tool and gently transferred into queen-rearing cups (JZBZ queen bee cells, HunterBee) with a small droplet of royal jelly. The cups were introduced to groups of bees in cages through a hole at the top of the cage, one cell per cage. For control, cups from the same grafting session were introduced to a queenless queen rearing foster colony containing six frames of bees and no brood. The queen cells were monitored to estimate the success rate of larval acceptance after 72 hours, and queen cell cup capping after ten days

## Queen pupa weight

Since the queen's weight changes during her adult life, we weighed queens at the pupa stage when the environment does not affect her weight. The queen pupae were gently removed from the q ueen cells on day 10 after grafting, approximately one to two days before emergence. The pupae were weighed on an analytical scale to the nearest milligram to estimate the queen's weight. Pupae that did not develop properly into queens such as those with distorted shapes, were excluded from weighing (less than 10% of the larvae).

**Experiment 1: The effect of the number of workers on queen development.** The cages were populated with different numbers of workers. The number of workers in each cage was estimated by weighing the day-old bees added to the cage, with an estimation of ten bees per gram. The cages were populated with 5 g, 10 g, 20 g, or 30 g of bees representing 50, 100, 200 or 300 bees respectively. Each treatment group included 19 or 20 replicates. One day old larva, placed in a queen cup, was introduced to each cage and tracked for development. The acceptance and development of the larvae, as well as the weight of the queens, were measured as detailed above. The workers used in the experiment were collected from two-parent colonies, and the larvae for grafting were obtained from a single-frame

**Experiment 2: The influence of larval age on queen development under lab conditions.** Larvae of three different ages were used for queen rearing in cages. The cages were populated with 20g of bees and fed *ad libitum* with honey and 90% pollen cake. To collect larvae of known age for grafting, a honey bee queen was caged in a queen excluder cage on a frame in a regular colony for 12 hours. The frame was checked for eggs at the end of the caging period, and the queen was released. The cage was kept on the frame for three days to ensure that the queen did not lay any more eggs on it. After three days, the frame was removed from the colony, and larvae aged 1–12 hours old, were used for queen rearing under controlled conditions. The frame was then returned to the queen excluder cage in the colony after about 45 minutes outside of the colony, kept under worm and humid conditions. A day later, on day four, the same frame was used for a second grafting of 24–36 hour-old larvae and returned to the colony. A final round of grafting was performed from the same frame for 48–60 hour-old larvae the next day. Each treatment group included 20 or 25 replicates. The acceptance and development of the larvae, as well as the weight of the pupae, were measured as detailed above. The workers used in the experiment were collected from two-parent colonies, and the larvae for grafting were obtained from a single-frame

**Experiment 3: The influence of pollen nutrition on queen development.** The defined richness of the pollen cake was varied between the experimental groups to test the effect of pollen nutrition on queen-rearing success. Each cage contained 200 bees and was fed pollen paste at four different levels of richness: Sugar water (60% sugar) with no pollen (P0%), 30% pollen mixed with 70% sugar water (P30%), 60% pollen mixed with 40% sugar water (P60%), 90% pollen mixed with 10% sugar water (P90%), all from the same pollen source. The cages were supplied with honey *ad libitum* as a carbohydrate source. Day-old larvae were grafted and introduced, one to each cage. Each treatment group included 25 replicates. The acceptance and development of the larvae, as well as the weight of the developing pupae, were measured as detailed above. The workers used in the experiment were collected from two-parent colonies, and the larvae for grafting were obtained from a single frame.

## Data analysis

A chi-square test of independence was used to compare larval acceptance and development rat. A post hoc analysis was performed using a chi-square test of independence for each pair with FDR correction for multiple comparisons. Queen's weight between treatment

groups was compared using an ANOVA test. The full data for the analysis can be found at S1 File. Data analyses for this study was conducted using the Real Statistics Resource Pack software for excel (Release 8.9.1). Copyright (2013 – 2023) Charles Zaiontz. www.real-sta-tistics.com.

## Results

### Experiment 1: The effect of the number of workers on queen development

We compared the queen-rearing success rate in groups of bees with different numbers of workers. The acceptance of the larvae was affected by the number of workers in the group. Groups of fifty workers (50W) accepted significantly fewer larvae, and none of the larvae developed to pupation compared to the other groups (Fig 1A). Groups of One hundred bees (100W) accept about half the larvae on day three and capped 20% (n = 20). Two hundred bees (200W) accepted 69% of the larvae and capped 48% (n = 19). Three hundred bees (300W) accepted 84% of the larvae and capped 63%. (Fig 1A, $\chi^2$ test for independence, $\chi^2_{(6)} = 24.2$, p < 0.001). The capping rate in the control foster colony was 66% (n = 20), which was not different from the 200W or 300W treatment groups. The larvae acceptance and pupation rate in the 50W treatment is significantly lower than that of 200W and the 300W treatment, and lower but not significant than the 100W ($\chi^2$ test with FDR correction, p < 0.05). The 100W treatment was not significantly different than the other treatments.

The weight of the queen pupa was affected by the group size. The pupae from the 100W treatment weighted 194 mg ± 1.0 (n = 2), queen pupae of the 200W treatment weight 228 mg ± 3.8 (n = 8), and queen pupae from the 300W treatment weight of 218 mg ± 2.6 (n = 10). The control foster colony raised pupae with an average weight of 236 mg ± 2.6 (n = 13). The differences between the treatments were significant (Fig 1B, One-way ANOVA, $F_{(3)} = 14.7$, p < 0.001). There are significant differences between the treatments groups were the 100W reared pupae are the lightest and the 300W treatment weight is lower than the foster colony raised pupae (Fig 1B, Tukey's post hoc test, p < 0.05).

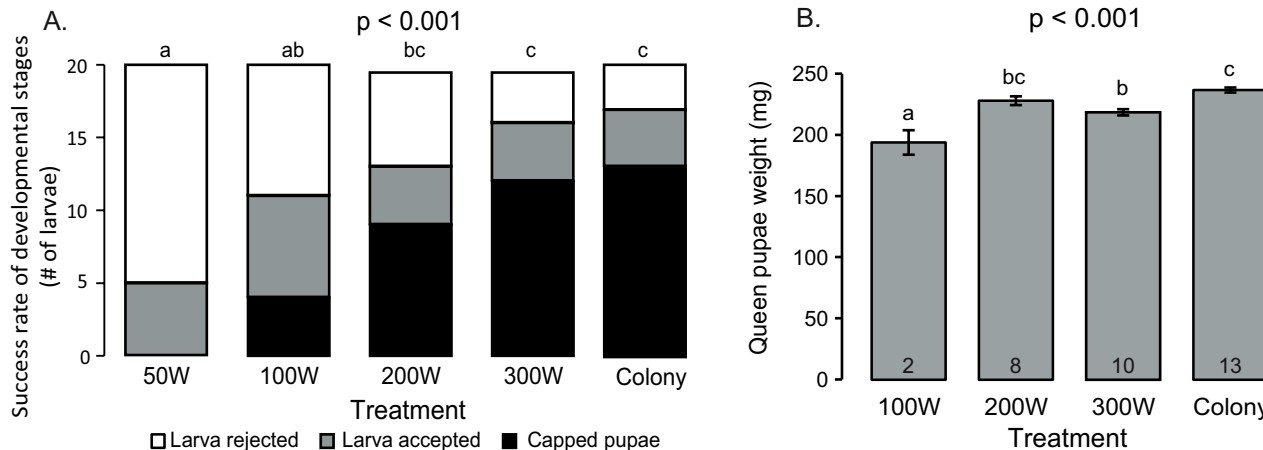

**Fig 1. The effect of the number of nurse bees on queen development.** Four treatment were compared: 50, 100, 200 and 300 workers and a control foster colony. **A.** Larvae development: The number of larvae rejected by the bees (White) accepted at day 3 (Grey) capped pupae (Black). Columns with different letters are significantly different in post hoc test ($\chi^2$ test with FDR correction p < 0.05). **B.** Mean ± SE of pupae weight; Columns with different letters are significantly different (One way ANOVA followed by Tukey pos-hoc test p < 0.05).

## Experiment 2: The influence of the larval age on queen development in cages

We tracked the acceptance and rearing success of queen larvae at three different larval ages used to grafting. The acceptance rate for young larvae at the age of 0–12 hours was 90%, where 75% of all larvae were capped (n = 20); Acceptance of 24–36 hours larvae were 96% and 68% were capped (n = 25). Only 24% of the 48–60-hour-old larvae were accepted and 16% were capped (n = 25). The age of the larvae has a significant effect on the success rate of the development to queens and the 48–60 h treatment group was lower in compare to the other treatments that had the same success rate (Fig 2A, Chi-square test for independence, $\chi^2_{(4)} = 37.3$, p < 0.001).

The weight of the queen pupae was affected by the age of the larvae. Queen pupae developed from 0–12 h larvae weighed on average 234 mg ± 5 (n = 14) while 24–36 h old weighed only 217 mg ± 6 (n = 15), the 48–60 h old larvae developed queens weighed only 206 mg ± 1 (n = 2). The difference between the treatment groups are significant (Fig 2B, One-way ANOVA, $F_{(2)} = 4.8$, $p = 0.016$). The pupae from the 0–12 h larvae were heavier than the two other treatment groups (Tukey post hoc test, $p = 0.03$).

## Experiment 3: The influence of pollen nutrition on queen development

Removing the pollen from the cages (P0%) resulted in zero acceptance of the larvae by the workers, and on day three, all the queen cups were empty and clean. Using P30% pollen paste, the bees accepted only a single larva (4%, n = 25). At medium pollen paste of P60%, the bees accepted 16% of the larvae (n = 25), and only two were capped (8%). Using the heavy P90% paste, 72% of the larvae were accepted by the workers on day three, and 64% were capped. The difference between the treatments is significant (Fig 3A, Chi-square test for independence, $\chi^2_{(6)} = 50.8$, $p < 0.001$). The P90% treatment was higher than all the other treatments ($\chi^2$ test with FDR correction, p < 0.05).

The average weight of the queen pupae in the P90% treatment was 237mg ± 4 (n = 13), in the P60% treatment it was 228mg ± 2 (n = 2), and the single pupa at the P30% treatment

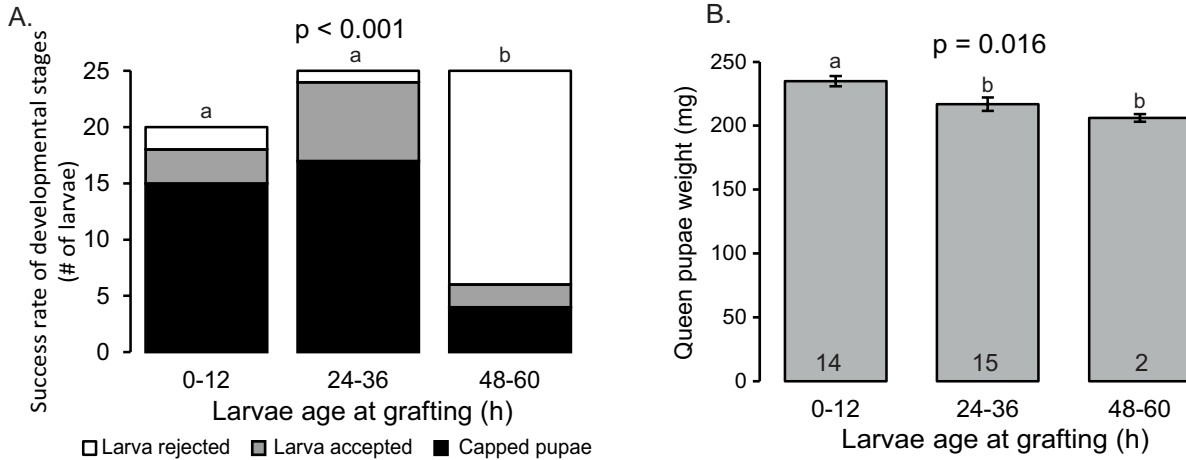

**Fig 2. The effect of larval age at grafting on queen development.** Three larvae age were compared 0–12; 24–36 and 48–60 hours **A.** Larvae development: The number of larvae rejected by the bees (White) accepted at day 3 (Grey) capped pupae (Black). Columns with different letters are significantly different in post hoc test ($\chi^2$ test with FDR correction p < 0.05). **B.** Mean ± SE of pupae weight; Columns with different letters are significantly different (One way ANOVA followed by Tukey pos-hoc test p < 0.05).

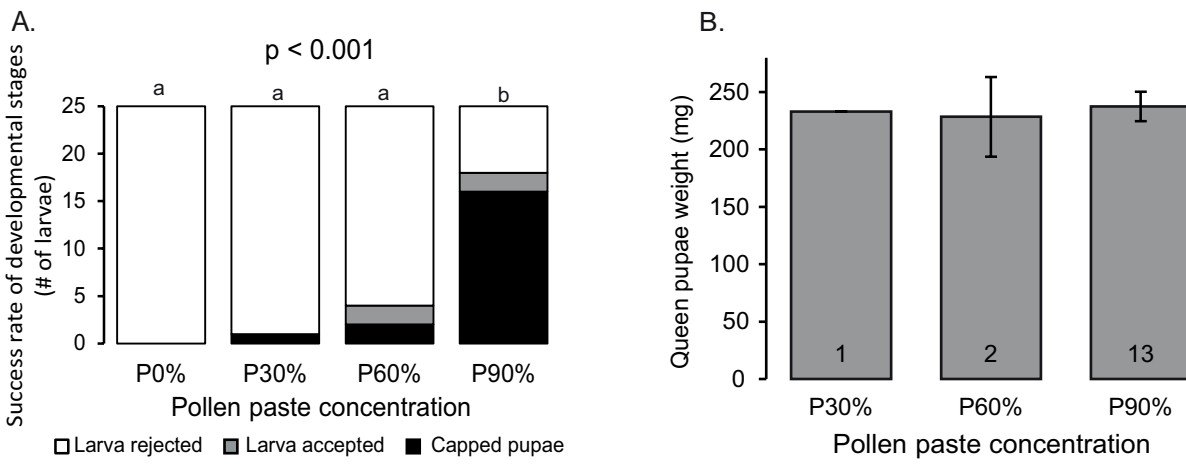

**Fig 3. The effect of pollen nutrition on queen development.** Four concentration of pollen were compared: 0%, 30%, 60% and 90%. **A.** Larvae development: The number of larvae rejected by the bees (White) accepted at day 3 (Grey) capped pupae (Black). Columns with different letters are significantly different in post hoc test ($\chi^2$ test with FDR correction $p < 0.05$). **B.** Mean ± SE of pupae weight. Due to the low success in queen rearing in the low pollen treatment the weight of pupa was not compared.

weight was 233mg (Fig 3B). Due to the low rearing rate of the queens in the low pollen groups, the differences between the groups were not compared.

## Discussion

The process of queen development depends on various biotic and abiotic factors [32–34]. Queen rearing is a social effort, and the workers' decision to rear a queen is made collectively [35]. In this study, we developed a new protocol to investigate the environmental and social conditions required for successful queen development in a semi-natural system. Our findings demonstrate that the workers' decision to rear a queen is complex and influenced by several social and physiological factors, including the number of bees in the group, their nutritional state, and the age of the larvae.

We found that a minimum number of worker bees is necessary to successfully rear a queen from a day-old larva. Fifty workers are not enough to rear a queen and one hundred bees do it very poorly. Groups of 200 or 300 workers succeeded in rearing queens in comparable success to a traditional to a foster colony in the current study, and consistent with data on success rate of queen rearing in foster colonies from other studies [3,36–38]. The wet weight of queen pupae reared in the lab by 200 bees was comparable to queens reared in foster colonies [13,39–43]. These findings suggests that 200 bees is the minimal reliable number required for successful queen rearing similar in weight to foster colonies. Since the rearing process of these queens differs from conventional methods, it would be valuable to investigate their performance in other parameters, such as mating success, colony introduction, and egg-laying capacity in the field.

The collective decision of the bees to participate in queen rearing is dynamic, with bees adjusting their behavior to changing conditions [35]. For example, eight bees can successfully rear a queen from a four-day-old larva [44], and even a single worker can care for a four-day-old larva [45]. However, fifty bees are insufficient to rear a queen from a one-day-old larva. Interestingly, in half of the cases in the current study, the bees began the rearing process and accepted the larva, but later abandoned it. This suggests that the decision to rear a queen depends on a complex interaction between the number of workers and the age of the larva.

How the workers assess their number in the cage is a puzzling question that should be further studied. Based on the findings of our first experiment, we continued to investigate the parameters affecting queen rearing in groups of two hundred bees, a practical and easily replicable method. As our protocol became more refined, our success rate in queen-rearing increased in the next experiments.

In our second experiment, we tested the effect of larval age on the quality of the developed queen using two hundred bees in each cage. Several studies have shown that the age of the larvae at grafting affects the quality of the developing queens. Younger larvae tend to develop into heavier and more fertile queens in traditional rearing in foster colonies [17,28,43,46,47] and in *in vitro* hand-rearing [34]. Our findings support these studies, as we found that larvae aged 0–12 hours developed into heavier queens compared to older larvae. We also demonstrated that the acceptance rate of larvae depends on their age, with larvae older than 48 hours rarely developing into queens [28]. The findings of this experiment provide a proof of concept for the new laboratory method of queen rearing using nurse bees, as we were able to consistently replicate former results from the literature.

The nutrition available to the workers also influences their decision to accept larvae for queen rearing. When we removed all pollen from the cages, the bees did not accept any of the larvae. A low-protein diet resulted in a low success rate in queen rearing, with only well-nourished workers accepting the task of queen rearing. This finding supports the hypothesis that the nutritional state of the bees is a crucial factor in their decision to rear a queen. The nutritional condition of the workers affects many functions of the bees, including the queen's egg-laying capacity [31], learning and memory [48] foraging behavior [49] as well as their physiology and immunity [50]. Our new method can also be used to test the effects of other nutritional factors on the success of queen rearing, such as different types of pollen, pollen supplements, and pesticide residuals [51–53].

Queen rearing is a major tool for bee stock selection and improvement. The lack of control over queen-rearing environmental conditions has turned this process into a "black box" where the breeder's influence on the queen's development has been limited. Rearing queens in the lab under controlled environment can help to study the factors that impact the workers behavior and the queen development. Moreover, isolating developing queens from the hive environment can reduce the impact of pathogens, such as the black queen cell virus, on the success of queen rearing [54]. This method can also be employed to test the effects of pesticides on honey bee health [26,55]. We anticipate that further studies using this new method will shed light on queen development and worker-rearing behavior.

## Supporting information

**S1 File. Rearing success table and pupae weight for experiments 1–3.**
(XLSX)

## Acknowledgments

We thank Dr. Victoria Soroker for help with the experiment design and the beekeeper Assaf Otmy for help with the bees.

## Author contributions

**Conceptualization:** Hagai Yehoshua Shpigler.

**Data curation:** Omer Kama.

**Formal analysis:** Omer Kama, Hagai Yehoshua Shpigler.

**Investigation:** Omer Kama, Hagai Yehoshua Shpigler.

**Methodology:** Omer Kama, Hagai Yehoshua Shpigler.

**Resources:** Hagai Yehoshua Shpigler.

**Supervision:** Hagai Yehoshua Shpigler.

**Writing – original draft:** Hagai Yehoshua Shpigler.

**Writing – review & editing:** Omer Kama.

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
