## [Decision Letter · Decision Letter 0]

10 Oct 2024

PONE-D-24-38555The Social and Nutritional Factors Controlling the Growth of Honey Bee (Apis mellifera) QueensPLOS ONE

Dear Dr. Shpigler,

Thank you for submitting your manuscript to PLOS ONE. After careful consideration, we feel that it has merit but the reviewers raise a number of important concerns. While I want to emphasize that publication in PLoS ONE does not require novelty, the methods need to be sufficiently described to allow exact replication, particularly for a methodology-based manuscript. It is also in your best interest to optimize the clarity of the presentation, as suggested by both reviewers.

We look forward to receiving your revised manuscript.

Kind regards,

Olav Rueppell

Academic Editor

PLOS ONE

Journal Requirements:

1. When submitting your revision, we need you to address these additional requirements. Please ensure that your manuscript meets PLOS ONE's style requirements, including those for file naming. The PLOS ONE style templates can be found at https://journals.plos.org/plosone/s/file?id=wjVg/PLOSOne_formatting_sample_main_body.pdf and https://journals.plos.org/plosone/s/file?id=ba62/PLOSOne_formatting_sample_title_authors_affiliations.pdf 2. In your Methods section, please provide additional information regarding the permits you obtained for the work. Please ensure you have included the full name of the authority that approved the field site access and, if no permits were required, a brief statement explaining why. 3. Please include captions for your Supporting Information files at the end of your manuscript, and update any in-text citations to match accordingly. Please see our Supporting Information guidelines for more information: http://journals.plos.org/plosone/s/supporting-information.

Reviewers' comments:

Reviewer's Responses to Questions

**Comments to the Author**

1. Is the manuscript technically sound, and do the data support the conclusions?

Reviewer #1: Partly

Reviewer #2: Partly

2. Has the statistical analysis been performed appropriately and rigorously? 

Reviewer #1: Yes

Reviewer #2: No

3. Have the authors made all data underlying the findings in their manuscript fully available?

Reviewer #1: No

Reviewer #2: Yes

4. Is the manuscript presented in an intelligible fashion and written in standard English?

Reviewer #1: Yes

Reviewer #2: Yes

5. Review Comments to the Author

Reviewer #1: This manuscript presents an interesting new method to rear queens using cages of worker bees in an incubator. The authors perform a series of experiments examining how factors known to affect queen rearing success in colonies affect rearing success in their system, demonstrating similar outcomes. It appears well conducted, and their conclusions are well supported. However, I would want to see a suggested next step for validating this as a viable queen rearing method. The next logical comparison would be to look at mating success, followed by mated queen performance. Aside from this, my strongest critiques relates to missing information in their methods section and some grammatical errors.

See below:

L40-41: This sentence is incomplete.

L42: Consider using the word “produce” instead of “grow.”

L57: I’m confused as to what the authors mean by “at emergence.” Emergence of what? Do you mean “in emergencies”?

L71: What makes a colony “superior” in this context?

L96: “This method allows researchers to conduct…”

L93-108: I don’t think the introduction is the best place to summarize the results. It’s somewhat redundant since you’ve already done this in the abstract. Consider eliminating mention of the final results.

L110: How many colonies were bees sourced from? Were the same colonies used for each experiment? How were the colonies maintained? How were they selected for use?

L122: I’m finding the authors’ description of ratios to be confusing (here and throughout). Consider using a different format to express this.

L133: “larval”

L143: “This manipulation did not affect the hatching success of the pupae.” How was this determined?

L154: I see this line after each experiment, and I don’t know what exactly it means. Two colonies were used for adult bees? For larvae? Did you repeat the experiment, once for each colony? How did you account for variation due to source colony in your experimental design and analysis?

L165: How long were the frames outside of the colony for? Could this have influenced the results?

L176: “larvae were”

L183: “The weight of the…”

L190: “and none of the larvae developed into…” I think this is a better way to express this.

L203-203: It seems like you should have had more queens than this to look at. Why not measure all of them?

L211: “Larval Age”

L213: “acceptance rate”

L223: “larvae”

L225: Is there a citation you can add after “collectively”?

Reviewer #2: Dear Authors,

Thank you for submitting your manuscript and endeavoring the peer-review process. I enjoyed reading your manuscript on a laboratory assay for honey bee queen rearing. I have now read your manuscript and reviewed the data provided in the supplementary materials. Your study reports on developing a laboratory rearing method for queen honey bees. You include an experiment where you asked the question, how many nurse bees are required to raise queen honey bees from larva in the lab; a second experiment where you adjust the age of the honey bee larva used in the queen cell grafting process in the lab; and a final experiment where you manipulate the proportion of pollen in the artificial diet used during the queen rearing process in the lab. I think it is interesting that you can rear queens in the lab, in semi-decent sample sizes, however, I was surprised to see that you consider this method novel, as rearing queen honey bees in the lab has been tried with success since the 80's (see Vandenberg and Shimanuki, 1987) and workers even earlier than the 80's. However, setting your work apart from the older works includes the use of nurse bees in the lab, which I think is an important distinction, however, you do not even mention these works, or the advantage of bringing worker bees into the lab for rearing, when rearing queens in the lab can be done without them. In fact, I would think bringing workers into the lab would be a lot of additional work, especially when queen rearing has been shown to be successful in their absence. The second experiment exploring the age of honey bee larvae used to graft queen cells is an exhausted study, also dating back to the 80's, and it's not clear why it was done in this study, other than to corroborate what beekeepers and researchers already know. The third experiment on the proportion of pollen used to rear queens also seems a bit corroborative. Perhaps including hypotheses and predictions would help elucidate the relevance and significance of these experiments to your overall study objectives will help?

Context of the Study and Study Objectives:

You've provided a nice review of queen honey bees, the breeding process and some of the significance behind the breeding process and their importance to apicultural and agricultural industries. However, you have not set up the gap, or need for a new laboratory rearing method for queens, especially when one already exists. There is a recent review of queen rearing methods Buchler et al., 2024, that may help supplement your background on current methods and practices of queen rearing, such that the introduction can be refocused to the gap this queen rearing method of yours fills. As I was reading the introduction, I got the sense that your objective was targeting an alternative queen rearing method in the lab to support queen rearing on a large scale to support the growing demand for queens in the industry, especially given the discussion around seasonality. But after reading your manuscript, this purpose seems unlikely and unattainable, as you still require continuous access to honey bee larvae and nurse bees in the field to rear queens in the lab. So I question the significance of rearing queens in the lab following your protocol with nurse bees because the context for alternative purposes or novel questions to address with such a method are not addressed or discussed.

Methods, Stats, and Reproducibility:

I've provided several annotations throughout the manuscript, please review for specific editorial suggestions, comments, confusions, and questions. In short, while I have a general understanding of the experiments, what you did, how you did them, the data you collected, and how it was analyzed and reported, I did not come to this understanding until the end of your results and figures. Your methods are not written straightforwardly, they do not provide details on sample size or replication, and are therefore not reproducible. I'm not sure why several chi square tests were done, when the majority of your data is 'presence/absence' binomial data that a binomial distribution should model well. After reviewing your data file in the supplementary attachment, they appear incomplete. Shared data requires more annotation. You've included your statistics, which is not required, but as is, is not complete. Treatments should be defined.

Reporting of Results:

Your results can be reported more concisely (see line-by-line comments). You essentially report all data in the text even though a figure is provided. I make suggestions on revising your figures, where you response variables are success rates of different developmental stages, instead of the current 'number of larvae'. Your discussion of results does not really discuss your results in the context of the field, aside from corroborating what is already known. Much of the discussion is a regurgitation of the results, as it is written in the results section.

I think with a clearer narrative on the importance of this method of queen rearing, including the questions and gaps of knowledge that could be addressed with such a method, this method would have merit.

Please consider my line-by-line annotations in the attached file, as there are several additional comments, suggestions, and questions.

All the best in the review process.

6. PLOS authors have the option to publish the peer review history of their article (what does this mean? ). If published, this will include your full peer review and any attached files.

**Do you want your identity to be public for this peer review?** For information about this choice, including consent withdrawal, please see our Privacy Policy .

Reviewer #1: No

Reviewer #2: No

---

## [Author Response · Author response to Decision Letter 0]

21 Jan 2025

Response to review

Hagai Y. Shpigler

Reviewer #1: This manuscript presents an interesting new method to rear queens using cages of worker bees in an incubator. The authors perform a series of experiments examining how factors known to affect queen rearing success in colonies affect rearing success in their system, demonstrating similar outcomes. It appears well conducted, and their conclusions are well supported.

However, I would want to see a suggested next step for validating this as a viable queen rearing method. The next logical comparison would be to look at mating success, followed by mated queen performance.

We thank the reviewer for their thoughtful and constructive feedback, as well as for recognizing the significance of our work. We fully agree with the reviewer’s suggestion regarding the next steps for validating this method, specifically evaluating mating success and mated queen performance.

We are working on these points in our lab and have added a paragraph to the Discussion section outlining these ideas as part of future research directions. This addition highlights the importance of comparing the outcomes of queens reared using our laboratory-based system to those reared traditionally, focusing on both mating success and the performance of mated queens in colonies under field conditions.

We believe this revision strengthens the manuscript by providing a clear roadmap for future work and addressing the reviewer’s insightful comments. Thank you once again for your valuable input.

Aside from this, my strongest critiques relates to missing information in their methods section and some grammatical errors.

See below:

L40-41: This sentence is incomplete.

We completed the sentence that read now: “Queen breeders, use specialized equipment and knowledge of honey bee biology to produce large number of queens”

L42: Consider using the word “produce” instead of “grow.”

We changed the wording as suggested

L57: I’m confused as to what the authors mean by “at emergence.” Emergence of what? Do you mean “in emergencies”?

We fixed the mistake as suggested

L71: What makes a colony “superior” in this context?

We change to highly productive colonies

L96: “This method allows researchers to conduct…”

We changed the sentence as suggested

L93-108: I don’t think the introduction is the best place to summarize the results. It’s somewhat redundant since you’ve already done this in the abstract. Consider eliminating mention of the final results.

We shortened and changed the paragraph as suggested.

L110: How many colonies were bees sourced from? Were the same colonies used for each experiment? How were the colonies maintained? How were they selected for use?

We added this data to the methods

L122: I’m finding the authors’ description of ratios to be confusing (here and throughout). Consider using a different format to express this.

We changed it to: the cages were supplied with pollen paste made of 90% pollen and 10% sugar water (60% sugar in the water)

L133: “larval”

Fixed

L143: “This manipulation did not affect the hatching success of the pupae.” How was this determined?

We removed this section from the study as we don’t have the full hatching data for all reared queens.

L154: I see this line after each experiment, and I don’t know what exactly it means. Two colonies were used for adult bees? For larvae? Did you repeat the experiment, once for each colony? How did you account for variation due to source colony in your experimental design and analysis?

The experiment was conducted using two parent colonies to provide the workers and a single frame of young larvae for all grafting procedures.

L165: How long were the frames outside of the colony for? Could this have influenced the results?

The frames were outside of the colony for approximately 45 minutes, during which they were covered with wet paper towels and kept in a humid room. We do not believe this short period influenced the results, as the larvae remained visibly viable throughout the process. Additionally, this handling time is consistent with standard practices in queen-rearing protocols, where larvae are routinely exposed to similar conditions without adverse effects.

L176: “larvae were”

Fixed

L183: “The weight of the…”

Fixed

L190: “and none of the larvae developed into…” I think this is a better way to express this.

Fixed

L203-203: It seems like you should have had more queens than this to look at. Why not measure all of them?

We appreciate the reviewer’s observation. As noted, we did not weigh pupae that exhibited poor development or had distorted shapes. A small number of such pupae were observed in each experiment, and we have now clarified this in the Methods section to ensure transparency. Including this explanation helps account for the discrepancy and ensures that our data represent only viable and properly developed pupae.

L211: “Larval Age”

Fixed

L213: “acceptance rate”

Fixed

L223: “larvae”

Fixed

L225: Is there a citation you can add after “collectively”?

Yes, We add the following reference:

Tarpy D. Collective decision-making during reproduction in social insects: a conceptual model for queen supersedure in honey bees (Apis mellifera). Curr Opin Insec Sci. 2024;66.

Reviewer #2: Dear Authors,

Thank you for submitting your manuscript and endeavoring the peer-review process. I enjoyed reading your manuscript on a laboratory assay for honey bee queen rearing. I have now read your manuscript and reviewed the data provided in the supplementary materials. Your study reports on developing a laboratory rearing method for queen honey bees. You include an experiment where you asked the question, how many nurse bees are required to raise queen honey bees from larva in the lab; a second experiment where you adjust the age of the honey bee larva used in the queen cell grafting process in the lab; and a final experiment where you manipulate the proportion of pollen in the artificial diet used during the queen rearing process in the lab. I think it is interesting that you can rear queens in the lab, in semi-decent sample sizes, however, I was surprised to see that you consider this method novel, as rearing queen honey bees in the lab has been tried with success since the 80's (see Vandenberg and Shimanuki, 1987) and workers even earlier than the 80's. However, setting your work apart from the older works includes the use of nurse bees in the lab, which I think is an important distinction, however, you do not even mention these works, or the advantage of bringing worker bees into the lab for rearing, when rearing queens in the lab can be done without them. In fact, I would think bringing workers into the lab would be a lot of additional work, especially when queen rearing has been shown to be successful in their absence. The second experiment exploring the age of honey bee larvae used to graft queen cells is an exhausted study, also dating back to the 80's, and it's not clear why it was done in this study, other than to corroborate what beekeepers and researchers already know. The third experiment on the proportion of pollen used to rear queens also seems a bit corroborative. Perhaps including hypotheses and predictions would help elucidate the relevance and significance of these experiments to your overall study objectives will help?

We appreciate the reviewer’s thoughtful comments and constructive feedback. We acknowledge the reviewer’s observation regarding the novelty of our method and agree that laboratory-based rearing of queens has been explored previously, such as in the work by Vandenberg and Shimanuki (1987). However, our innovation lies in the incorporation of nurse bees into the laboratory setting to rear queens, rather than a completely in vitro method.

To address this, we have added a paragraph to the Introduction discussing previous work on in vitro larval rearing of honey bees. This addition clarifies how our method builds on prior research and highlights the advantages of using nurse bees in the lab. While in vitro queen rearing provides a fully artificial method, it does not allow for the study of the nursing behavior of worker bees, which is integral to understanding queen growth. By incorporating nurse bees into a controlled laboratory environment, our method provides a semi-natural system that enables the study of queen development alongside worker behavior, offering unique opportunities for both research and potential improvements in queen rearing.

We respectfully disagree that bringing nurse bees into the lab is a particularly challenging process. In our study, we used one-day-old bees, which are easy to collect and handle. Furthermore, we used known principles, such as the effect of larval age on queen weight, as a proof of concept for validating our method. While these findings corroborate existing knowledge, they also demonstrate the reliability of our new system.

Lastly, we suggest that our method not only provides insights into queen growth but also holds potential for improving queen rearing practices. We believe these clarifications and the additional context enhance the manuscript and address the reviewer’s concerns. Thank you for highlighting these important points.

Context of the Study and Study Objectives:

You've provided a nice review of queen honey bees, the breeding process and some of the significance behind the breeding process and their importance to apicultural and agricultural industries. However, you have not set up the gap, or need for a new laboratory rearing method for queens, especially when one already exists. There is a recent review of queen rearing methods Buchler et al., 2024, that may help supplement your background on current methods and practices of queen rearing, such that the introduction can be refocused to the gap this queen rearing method of yours fills.

We appreciate the reviewer’s insightful comments regarding the context and objectives of the study, as well as the suggestion to strengthen the introduction by addressing the gap our method fills. Thank you for referencing the recent review by Büchler et al. (2024), we did cite the older version of this paper from 2013. We have incorporated this new reference into the Introduction and used it to better contextualize our work within the landscape of existing queen-rearing methods.

In the revised introduction, we have explicitly highlighted the limitations of current queen-rearing methods, including fully in vitro systems, which do not allow for the study of nursing behavior or the semi-natural processes involved in queen development. We also clarified how our method addresses these gaps by providing a novel approach to studying queen rearing in a controlled environment while maintaining the influence of worker bee behavior.

We believe these revisions better align the manuscript with the reviewer’s suggestions and improve the clarity of our study’s objectives and significance. Thank you again for this valuable feedback.

As I was reading the introduction, I got the sense that your objective was targeting an alternative queen rearing method in the lab to support queen rearing on a large scale to support the growing demand for queens in the industry, especially given the discussion around seasonality. But after reading your manuscript, this purpose seems unlikely and unattainable, as you still require continuous access to honey bee larvae and nurse bees in the field to rear queens in the lab. So I question the significance of rearing queens in the lab following your protocol with nurse bees because the context for alternative purposes or novel questions to address with such a method are not addressed or discussed.

We thank the reviewer for their thoughtful comments and for highlighting the need to refine the context and aims of our manuscript. We agree that the original framing of our method may have implied broader applications for large-scale industrial queen rearing, which could be misleading given the current dependency on continuous access to honey bee larvae and nurse bees.

To address this, we have revised the Introduction and Discussion sections to present the aims and significance of our study in a more modest and research-focused context. Specifically, we emphasize that our protocol is designed primarily as a tool for studying queen rearing and development under controlled conditions, rather than as a scalable industrial solution. Additionally, we have discussed how this method can be used to explore novel research questions related to queen growth, nursing behavior, and environmental factors affecting queen development, which are difficult to investigate using traditional in-field methods.

We believe these revisions better align the manuscript with the reviewer’s observations and enhance the clarity and relevance of the study's objectives. Thank you again for this valuable feedback.

Methods, Stats, and Reproducibility:

I've provided several annotations throughout the manuscript, please review for specific editorial suggestions, comments, confusions, and questions. In short, while I have a general understanding of the experiments, what you did, how you did them, the data you collected, and how it was analyzed and reported, I did not come to this understanding until the end of your results and figures. Your methods are not written straightforwardly, they do not provide details on sample size or replication, and are therefore not reproducible. I'm not sure why several chi square tests were done, when the majority of your data is 'presence/absence' binomial data that a binomial distribution should model well. After reviewing your data file in the supplementary attachment, they appear incomplete. Shared data requires more annotation. You've included your statistics, which is not required, but as is, is not complete. Treatments should be defined.

We thank the reviewer for their detailed and thorough review of our manuscript. In response to the reviewer’s feedback, we have revised the Methods section to include more detailed descriptions of sample size, replication, and experimental procedures to ensure clarity and reproducibility. We also reviewed and addressed all annotations provided throughout the manuscript, incorporating the majority of the reviewer’s suggestions into the revised version.

Regarding the supplementary data, we have added annotations and clarified the treatments to support the analysis and improve the usability of the shared data. This ensures the supplementary materials are more informative and accessible to readers.

We also acknowledge the reviewer’s point about the statistical tests used. After careful consideration, we revisited our analysis and explored the suggested approach. However, as the results were consistent with those obtained using the chi-square tests, we have opted to retain our original analysis. We believe this approach remains appropriate for our data while ensuring consistency with the reported results.

Thank you for your thoughtful feedback, which has helped us improve the clarity and rigor of the manuscript.

We believe these changes address the reviewer’s concerns and significantly improve the manuscript. Thank you again for your valuable feedback.

Reporting of Results:

Your results can be reported more concisely (see line-by-line comments). You essentially report all data in the text even though a figure is provided. I make suggestions on revising your figures, where you response variables are success rates of different developmental stages, instead of the current 'number of larvae'. Your discussion of results does not really discuss your results in the context of the field, aside from corroborating what is already known. Much of the discussion is a regurgitation of the results, as it is written in the results section.

We greatly appreciate the reviewer’s detailed comments and suggestions regarding the reporting of results and discussion. In response to your feedback, we have made significant revisions to the Results section, condensing the text to avoid too much redundancy and ensuring that data presented in the figures are not repeated unnecessarily in the main text. Additionally, we revised several figures based on your suggestions to better align with the success rates of dev

---

## [Editor Report · Decision Letter 1]

24 Jan 2025

Social and nutritional factors controlling the growth of honey bee (Apis mellifera) queens

PONE-D-24-38555R1

Dear Dr. Shpigler,

We’re pleased to inform you that your manuscript has been judged scientifically suitable for publication and will be formally accepted for publication once it meets all outstanding technical requirements.

Kind regards,

Olav Rueppell

Academic Editor

PLOS ONE
---

## [Editor Report · Acceptance letter]

PONE-D-24-38555R1

PLOS ONE

Dear Dr. Shpigler,

I'm pleased to inform you that your manuscript has been deemed suitable for publication in PLOS ONE. Congratulations! Your manuscript is now being handed over to our production team.

Kind regards,

on behalf of

Dr. Olav Rueppell

Academic Editor

PLOS ONE